# A Research on Result Interpretability of Medical AI Based on Large Language Model

## Abstract

Explainability is one of the important challenges facing the application of medical AI. The existing AI explainability research is more of a kind of process explainability study. Drawing on the behavioral habits of human beings to communicate on a certain topic, this paper proposes a definition of result interpretability for medical AI, divides explainable medical AI research into three phases: data explainability, process explainability and result interpretability, and argues that once an AI model reaches a certain result interpretability metric, we can accept its conclusions and apply it to the clinic without having to wait until human beings fully understand the operation and decision-making mechanism of the AI model before using it. In this regard, we propose the concept of interpretative integrity. Further, we propose an architecture for result-interpretable medical AI system based on AI-Agent and build a result-interpretable system around risk prediction AI model for amyloidosis, which enables professional interpretation of the result of the risk prediction model for amyloidosis disease through a large language model and supports professional Q&A with clinicians. The implementation of the system enhances clinicians' professional acceptance of medical AI models, and provides a more feasible realization path for the large-scale application of medical AI-assisted diagnosis.

## 1 Introduction

Machine learning and deep learning are increasingly used in healthcare, yet these AI models are typically black-box in nature. Relying on unexplained black-box models to make decisions may lead to blindness in clinical diagnosis and treatment, which leads to relatively low acceptance of medical AI by clinicians Varghese (2020); Taylor & Fenner (2019). Explainable Artificial Intelligence (XAI) plays a crucial role in promoting human understanding and trust in deep learning systems. Many scholars have attempted to endow AI with explainable capabilities from a variety of perspectives, including data, neural network structure, and algorithm design. The goal of XAI is to reduce application risk by providing explanations for black-box model decisions or making the model decision-making process transparent Swamy (2023); Kim et al. (2023). The existing researches on XAI are complex and diverse, but in general they can be categorized according to the scope of the concern, the specific methodology and the implementation Das et al. (2020); Madsen et al. (2022); Danilevsky et al. (2020); Sarti et al. (2023); Enguehard (2023); Yin et al. (2022); Wu et al. (2023).

Overall, the existing explainability research of AI is a kind of study from the perspective of model and algorithm designers, which is still difficult to be accepted by the applicators. At the same time, the development of AI has far exceeded the progress of existing explainability studies. So, one of the realistic questions we are faced with is under what circumstances can humans be truly confident in using AI in specialized areas such as healthcare? Can this be achieved by giving AI algorithms and models a threshold standard for precision and accuracy? Or should humans use AI after the black-box effects of AI models are fully understandable, reasonable, and controllable?

Compared with other fields, medical treatment puts forward higher requirements for theories, technologies and methods related to explainable AI. Explainability has become a key issue that must be faced by the clinical application of medical AI, and many scholars have conducted research on this issue. Singh et al. (2020) and Salahuddin et al. (2022) focused on a review of XAI applications

in medical image analysis. Antoniadi et al. (2021) provided a review of XAI for clinical decision support (CDS). Payrovnaziri et al. (2020) provided a review of XAI for electronic health records (EHR). In general, the most commonly used XAI methods in the medical field today are SHapley Additive exPlanations (SHAP) Lundberg et al. (2017), Local Interpretable Model-agnostic Explanations (LIME) Ribeiro et al. (2016); Dave et al. (2020) and Gradient-weighted Class Activation Mapping (GradCAM) Zhou et al. (2015). From the above analysis, the existing explainable medical AI research belongs to the application of general XAI methods in medical treatment, which has not yet fully considered the professional requirements in clinical scenarios, and is still unable to solve the problem of medical AI results being adopted and directly used by physicians.

Rapid development of large models provides new ideas for research on the explainability of medical AI Yunxiang et al. (2023); LUO et al. (2022); WU et al. (2023); XU et al. (2023); YANG et al. (2023), and we can utilize the natural language interaction and content generation capabilities of large language models(LLMs) to provide professional interpretations of the results of traditional medical AI models and algorithms, which is more conducive to the direct application of medical AI in the clinic. To address this problem, this paper proposes the concept of result interpretability, which provide a new idea for explainable medical AI, i.e., instead of letting doctors believe in AI through the describability of AI's reasoning process, letting doctors accept AI's conclusions through the professional interpretation of the results.

## 2 RESULT INTERPRETABILITY AND ITS METRICS OF MEDICAL AI

### 2.1 PROCESS EXPLAINABILITY AND RESULT INTERPRETABILITY OF MEDICAL AI

Various prior publications debate the nuances in defining explainability of neural networks Dosilovic et al. (2018); Chakraborty et al. (2017). Despite the differences in definitions of XAI, the essence of these definitions is to help the user to have a clear understanding of the model's decision-making process in a simple and clear way, and then to trust the model's results. This kind of explainability is concerned in fact with the describability of the internal structure of the model and the computational process. Thus, it can be defined as process explainability. Medical AI's process explainability is essentially a kind of process descriptability that aims to increase human trust in the AI and thus accept the conclusions made by the AI. However, the rapid expansion of the parameter scale of AI models and the limitations of human cognitive level will lead to the long-term problem of cognitive alignment between AI and humans, and the process explainability cannot be realized in a short period of time, which in turn will affect the large-scale application of medical AI.

Research on process explainability of medical AI is more often used to guide the design and optimization of models for better results. However, what professionals need is a full interpretation of the AI conclusions from a professional perspective, not an explanation from the model builder of how the model was constructed and works. For this reason, we defines the result interpretability of medical AI from the perspective of the clinician as a user of the AI algorithm, that is, human understanding of the specialized knowledge, decision rationale and causal relationships underlying the decision/prediction results of AI models. Result interpretability is analogous to the fact that when humans explain their behavior, it is impossible to explain how the brain works or how neurons conduct, but humans can explain their behavior through knowledge and experience in a way that is understandable and acceptable to the audience.

Process-explainable medical AI attempts to allow physicians to understand the training and reasoning process of medical AI, but is divorced from the physician's area of expertise. Despite increasing physician's trust in the AI, the lack of medical expertise and judgmental logic still greatly limits its practical application. For example, when a patient asks a physician why he or she may be suffering from a certain disease, the physician cannot say that it is because the AI model has made such a prediction. Physicians still need to make professional analysis and judgment based on clinical test indicators and relevant findings. Considering this, we redefines the different phases of explainable medical AI research and categorizes them into three phases: data explainability, process explainability, and result interpretability, as shown in the Fig.1. The inclusion of result interpretability allows explainable medical AI research to cover the whole process from training to deployment and then from inference to clinical application, which will greatly facilitate the application of medical AI in real clinical scenarios.

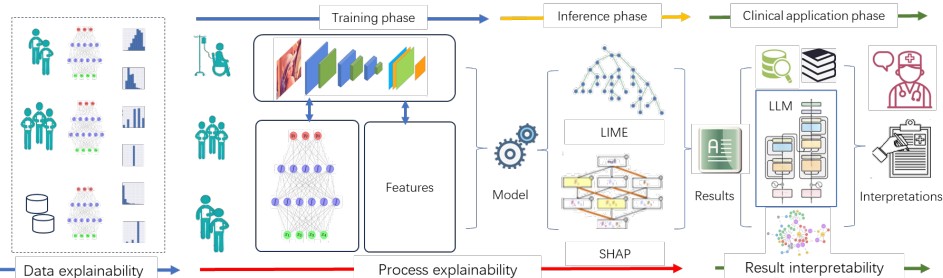

Figure 1: Data explainability, process explainability and result interpretability

## 2.2 INTERPRETATIVE INTEGRITY AND ASSESSMENT

To enable result-interpretable medical AI models or algorithms to be used effectively in the clinic, the first issue to be addressed is how to judge the adequacy of the interpretation of AI results. That is, what situation or condition is reached where we believe the interpretation of the results of the AI model or algorithm has been accomplished from a professional point of view and is able to be accepted by the physician and applied in the clinic. Thus, it becomes necessary to determine the basis for judging the adequacy of the interpretation of medical AI results. In other words, once the professional interpretation of the results of a certain AI algorithm meets the evaluation criteria of adequacy, we can consider it to be able to be understood and accepted by physicians from a professional point of view and meet the requirements for its application in the professional field without having to wait until human beings fully understand the internal operation and decision-making mechanisms of the AI black box model before using it. This criterion of judgment is more in line with the behavioral habits of human beings in communicating on a certain topic, i.e., if the other party gives a reasonable interpretation, we accept his suggestions or conclusions, rather than deciding whether to accept his suggestions or conclusions on the basis of how he thinks about them.

The development of LLM allows machines and humans to communicate fluently and expertly using natural language, which also provides a technical means to realize results-interpretable medical AI. Therefore, we propose the concept of interpretative integrity for result-interpretable medical AI system, it reflects whether the system meets the basic requirements for professional communication, it also determines whether a result-interpretable medical AI system can be truly embedded in the diagnosis and treatment process and accepted by clinicians. Interpretative integrity includes three dimensional indicators: consistency, coverage and professionalism. **Consistency** is the degree to which the interpretation matches the contextually relevant knowledge, emphasizing the relevance of the interpreted content, require that the content of the interpretation fully considers and incorporates the specific input information and relevant knowledge on which the medical AI results are based, such as patient clinical test indicators and medical records. **Coverage** refers to whether the content of the interpretation comprehensively covers the clinical concerns, and respond to the questions posed by clinicians with clear interpretations or answers. Coverage emphasizes the comprehensiveness of answering questions and interpretations, which helps clinicians progressively understand and accept medical AI results in depth from a medical professional perspective.**Professionalism** refers to the qualities of the result interpretation that are consistent with domain knowledge, experience and standardized terminology, and emphasizes that the content of the interpretation should not only be accurate and reliable, but also have professional insights and linguistic normativity in medical field. The professionalism of the interpreted content of medical AI results currently still needs to be measured by specialized physicians.

In addition to the three metrics involved in interpretative integrity, there are other functional requirements for system design that need to be fully considered. These requirements are fundamental guarantee that the result-interpretable medical AI system can be practically applied in the clinic. However, the realization of these requirements cannot rely solely on the content generation capabilities of a LLM, and requires that the result-interpretable medical AI system have a flexible system architecture and a dynamic combination of functions. Therefore, in this paper, we use AI-Agent XI et al. (2023); LIN et al. (2023); HUANG et al. (2023) to build result-interpretable medical AI system.

# 3 A RESULT-INTERPRETABLE RISK PREDICTION SYSTEM FOR AMYLOIDOSIS (**RIP4LCA**)

To validate the effectiveness of the result-interpretable approach, we designed a clinically oriented result-interpretable disease risk prediction system for amyloidosis, a rare hematological disease. The system not only predicts a patient's risk of suffering amyloidosis, but also provides an expertise-based diagnostic interpretation of the results. At the same time, the system utilizes interpretative integrity assessment tools to ensure the quality of the interpretation content.

Primary light chain (AL) amyloidosis is a rare hematological disease with multi-organ involvement that is associated with high mortality and difficult early diagnosis. Nearly one-third of patients with amyloidosis experience five or more consultations before diagnosis, which may lead to a poor prognosis due to delayed diagnosis, with up to 30% of patients with AL amyloidosis dying within the first year of diagnosis. For this disease, we designed and realized an ensemble learning risk prediction algorithm for amyloidosis with high accuracy (>90%) for early disease risk prediction, using routine screening indicators (gender, age, routine blood tests, urine test results, biochemical results, and echocardiography results as reference factors). Due to the professional requirements of clinical diagnosis, the high accuracy of AI prediction algorithms does not fully meet the requirements of clinical application of assisted diagnostic systems, and professional interpretation and diagnostic basis need to be provided as a reference to ensure that physicians can understand the conclusions of the AI algorithms from a professional perspective. For this reason, we constructed a result-interpretable amyloidosis risk prediction system (as shown in Fig.2), and realized multi-round dialogue and professional Q&A.

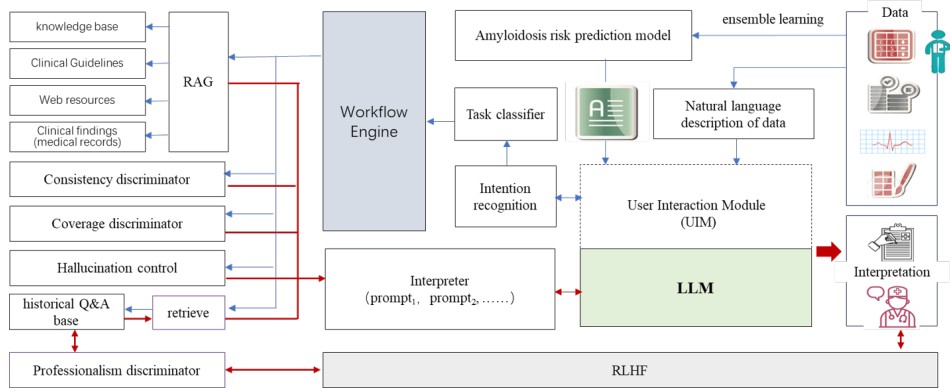

Figure 2: Result-interpretable risk prediction system for amyloidosis

## 3.1 INTERPRETATION OF AMYLOIDOSIS RISK PREDICTION

When the risk prediction model for amyloidosis gives a prediction result for a sample, knowledge retrieval related to the patient's clinical exam data is first performed through the AI-Agent workflow engine using RAG. Based on the features used by the risk prediction model, the relevant knowledge triad is queried in the knowledge graph and converted into an individual-related clinical knowledge base for a specific case in natural language form. Multiple triads of the same entity and relationship type can be converted into one piece of knowledge. For example, (Amyloidosis, Clinical manifestations, Hypertrophy of the tongue) and (Amyloidosis, Clinical manifestations, Periorbital purpura) are converted to "Clinical manifestations of amyloidosis include symptoms such as hypertrophy of the tongue and periorbital purpura". Then, the prediction results of the amyloidosis risk prediction model are input to the interpreter along with the case knowledge, the interpreter generates specialized interpretations of amyloidosis prediction results using LLM, and the consistency calculation module is used to complete the consistency assessment of the interpretation. Interpretations with highest consistency are presented to clinicians along with amyloidosis risk prediction results and model inputs. The basic flow is shown in Fig.3.

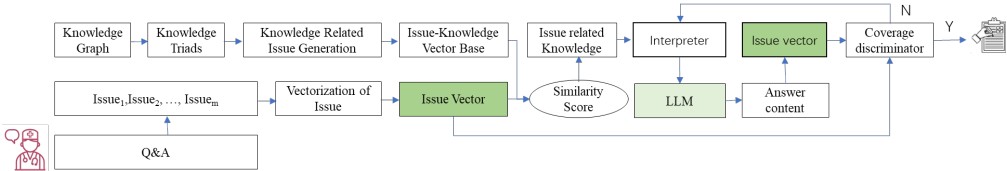

Figure 3: Interpretation of amyloidosis risk prediction

The consistency of interpretation is basically calculated as follows: First, the knowledge in the individual-related clinical knowledge base is sliced into text fragments according to a fixed maximum length and semantic separation rule, form the knowledge slices set $K = \{k_1, k_2, \cdots, k_n\}$. To avoid the effect of the order of the text in the prompts on the interpretation generation, for an interpretive task, we randomly sort the knowledge slices set $K$ several times and construct it as a prompt set $P = \{p_1, p_2, \cdots, p_m\}$ using prompt templates,

$$p_i = Prompt(shuffle(K, i)), i \in \{1, 2, \cdots, m\} \tag{1}$$

Where, $shuffle(K, i)$ denotes the $i$th randomized sorting operation on the knowledge slices set $K$. Each $p_i$ in the prompt set $P$ is entered independently into the LLM to generate interpretation content:

$$O = \{o_i | o_i = LLM(p_i)\}, i \in \{1, 2, \cdots, m\} \tag{2}$$

To assess whether the content of the interpretation is consistent with factual knowledge and maximizes the inclusion of the necessary factual knowledge, we perform item-by-item semantic similarity matching between interpretations and the set $K$ of knowledge to select the interpretation with the optimal degree of consistency. Specifically, we use the sum of the cosine similarity of $o_i$ in the set $O$ of interpretation to each $k_i$ in set $K$ of knowledge as the consistency score of the current interpretation. In the end, we choose the generated content with the highest consistency score and output it as the contents of the interpretation.

$$s_i = \sum_{j=1}^{n} \frac{o_i \cdot k_j}{\parallel o_i \parallel \parallel k_j \parallel}, o_i \in O \tag{3}$$

## 3.2 Q&A AND INTERACTION

In addition to interpretive analysis of the results, the system provides professional Q&A and interaction for physicians to give further analytical assistance. The process flow of Q&A is shown in Fig.4.

Figure 4: Process of Q&A

First, we convert the disease-specific knowledge graph of amyloidosis into a natural language form of knowledge base $K$ and the knowledge in the base is transformed from the triad in the knowledge graph. Converting knowledge from ternary form to natural language form allows LLM to understand the knowledge more effectively. After converting all the triads in the knowledge graph into natural language knowledge, we can generate one or more issue for each knowledge slice. Finally, the issue

in the issue-knowledge base is vectorized using a semantic vectorized pre-training model, and the generated embedding representation of the issue is stored in the issue-knowledge vector base $M$.

When a physician asks a question, we define the physician's question as $Q_D$. We first slice the $Q_D$ into multiple segments with independent semantics, and generate corresponding issue for each semantics slice to construct the issue set $I_D$, so that $Q_D$ can be represented as the issue set of all semantic slices. We retrieve relevant knowledge for the physician's question by finding similar issues in the issue-knowledge vector base $M$ for each issue in $I_D$. First, we vectorize all issues in $I_D$ using the same semantic vectorization pre-training model as used to construct the issue-knowledge vector base, and with the vector similarity calculation, we are able to retrieve the most relevant knowledge slice for each issue $i_D$ in $I_D$ from the issue-knowledge vector base $M$. To ensure the effectiveness and comprehensiveness of knowledge retrieval, we refer to the voting mechanism in KnowledgeNavigatorGUO et al. (2024) to assign a composite similarity score to each knowledge slice $k$ corresponding to similar issues $M_k$ using the sum of vector similarity score between the similar issues $M_k$ generated from $k$ and $i_D$ from the physician's question:

$$Score(M_K, i_D) = \sum_{m \in M_K} Sim(i_D, m) \tag{4}$$

Then, we retrieve knowledge slice with the highest similarity scores $S$ as the relevant knowledge corresponding to $i_D$. Finally, we summarize the knowledge involved in each issue in $I_D$ and use it as a prompt to generate a preliminary answer $C$ to the physician's question through LLM.

$$S = Top\Big\{Score(M_K, i_D) \mid k \in K, i_D \in I_D\Big\} \tag{5}$$

To ensure the comprehensiveness of the content of the answers generated in response to physician's questions, we assess the quality of the preliminary answers by performing a coverage calculation, and propose a coverage calculation method of multi-issue similarity comparison.

First, we semantically slice $C$ to construct the set $T$ of semantic slices of the answer content, and utilize the LLM to generate 5 semantically similar issues with each semantic slice $T_i \in T$ as the target answer, respectively, so as to construct the issue set $Q_C$ of the answer content. Then, we perform a one-to-one similarity comparison between the issue set $I_D$, which represents the physician's original issue, and the reconstructed answer issue set $Q_C$. Finally, we define the similarity scoring function $S$ and evaluate the coverage by calculating the ratio of the number of issues in $I_D$ that can find at least one similar issue in $Q_C$ to the total number of issues in $I_D$:

$$N = \sum_{i \in I_D} \sum_{q \in Q_C} \mathbb{I}\Big(S(i, q) \geq \theta\Big) \tag{6}$$

$$Coverage = \frac{N}{\mid I_D \mid} \tag{7}$$

where $I(S \geq \theta)$ is an indicator function, and the value of the indicator function is 1 when the condition is satisfied and 0 otherwise. For this indicator function, we set the similarity score threshold to $\theta$, i.e., two issues are considered to be the same issue when their similarity scores are greater than $\theta$. The system will output to the clinicians the content of the responses whose coverage meets the set threshold or reaches a certain number of judgment rounds.

## 4 EXPERIMENTATION AND ANALYSIS

### 4.1 ASSESSMENT OF THE VALIDITY OF RIP4LCA SYSTEM

We conducted experiments on the system realized in this paper around three measures of interpretative integrity, namely consistency, coverage and professionalism. In order to minimize the interference of different LLM, we used multiple LLMs for comparison in each group of experiments. Two groups of experiments are designed to assess the effectiveness of the system. For each group of

experiments, we designed the same 380 interpretative tasks and 200 Q&A-type tasks, which were derived from clinical practice. Each group of experiments required professional interpretations of amyloidosis risk prediction results and related inputs, also required professional answers to questions. For the first group of experiments, we directly invoked the LLM without using any knowledge engineering or specialized information supplements, and simply used the generalized capabilities of the LLM itself to finish all interpretative tasks and Q&A-type tasks. The second group of experiments used RIP4LCA system realized in this paper to provide professional interpretations of the amyloidosis risk prediction results and related input indicators, as well as professional answers to physician's questions.

For the interpretative tasks in each group of experiments, we computed the degree of consistency separately; for the Q&A-type tasks in each group of experiments, we computed the degree of coverage separately, in which the threshold of $\theta$ for the indicator function $I(S \geq \theta)$ in the process of coverage computation was set to 0.80. For all interpretive tasks and Q&A-type tasks in each group of experiments, we commissioned the clinical professionals to conduct a measure of professionalism. The measure of professionalism was scored on a scale of 1 to 5, with a maximum of 5 and a minimum of 1 for content professionalism, given independently by 6 different clinicians, and the average of each clinician's score was taken. The results of the consistency, coverage and professionalism evaluations of each group of experiments are shown in Table.1.

Table 1: Assessment of the validity of RIP4LCA system

| Model | Group1(LLM directly) | | | Group2(RIP4LCA system) | | |
|---|---|---|---|---|---|---|
| | Consistency | Coverage | Professionalism | Consistency | Coverage | Professionalism |
| Qwen2-72B | 0.66 | 0.44 | 3.22 | 0.77 | 0.84 | 4.27 |
| Qwen1.5-32B | 0.62 | 0.46 | 3.21 | 0.76 | 0.77 | 4.19 |
| GPT-3.5turbo | 0.59 | 0.33 | 2.77 | 0.74 | 0.62 | 3.99 |

Regardless of which LLM was used, the first group of experiments performed poorly on the consistency, coverage and professionalism. Specifically, in terms of consistency, the output content did not adequately use the expertise that underlie the clinical diagnosis and treatment of amyloidosis; in terms of coverage, the output content did not adequately cover all the concerns of the questions posed by the users, and answered the questions inappropriately or only responded to a part of the concerns; and in terms of professionalism , content output using the three LLMs scored only 3.22, 3.21 and 2.77, respectively, as assessed by physicians. This is attributed to the fact that the LLMs lacked specialized knowledge related to rare hematological diseases and could not meet the requirements of clinical diagnosis and treatment. It also suggests that the LLM alone cannot achieve a result-interpretable medical AI system.

In the second group of experiments, we use RIP4LCA system realized in this paper to complete the test, and the system improves the quality of the output by invoking the corresponding tools through the AI-Agent to complete the expertise supplementation, consistency discrimination and coverage discrimination. For the interpretive task, the system adopts the cosine similarity algorithm to retrieve the relevant expertise, and shuffle the retrieved relevant knowledge to generate the interpretation content through the system's interpreter by invoking the LLM, and then selects the optimal interpretation content for the output by using the consistency discrimination, and the number of shuffle times was selected to be 20 in the experiment. As can be seen from Table 1, the system realized in this paper has a substantial improvement with 77%, 76% and 74% consistency on the three LLMs, respectively. For the Q&A-type task, the system adopts the knowledge retrieval enhancement based on multi-issue decomposition and a coverage calculation method of multi-issue similarity comparison, and selects the optimal answer output through the coverage discriminator, thus, it also performs well in the Q&A-type task. Finally, in terms of professionalism, no matter which LLM is used, the RIP4LCA system scores high in both interpretive and Q&A-type tasks.

As can be seen from the results of the above two groups of experiments, the result-interpretable amyloidosis risk prediction system realized in this paper is able to provide clinicians with profes-

sional, clinically realistic interpretations and answers based on the results of the medical AI and the relevant input indicators, which improves the ability to deploy medical AI in clinical diagnosis and treatment.

## 4.2 INFLUENCE OF KNOWLEDGE REPRESENTATION ON CONSISTENCY OF INTERPRETATION

For the result-interpretable amyloidosis risk prediction system, we further experimentally analyze the impact of the knowledge representation in natural language proposed in this paper on the consistency of interpretation. Experiments were conducted to evaluate the effect of different knowledge representations on the consistency of the interpretation by using three knowledge representations to implement expertise supplementation respectively.

We conducted three groups of experiments, each containing 380 interpretive tasks, and took the average consistency of the output content of these tasks as the result of that group of experiments. For the first group of experiments, we use the knowledge representation of the ternary, without any processing or transformation, as an external knowledge supplement, and generate the interpretation through the system's interpreter by invoking LLM. For the second group of experiments, we used triads with merged head and tail entities as external knowledge supplements for interpretative content generation. For the third group of experiments, we used natural language knowledge representation, i.e., we transformed the original knowledge triad into a natural language knowledge text, which was used as knowledge supplements for interpretative content generation. Each group of experiments was conducted using three LLMs and the results are shown in Table.2.

Table 2: Influence of knowledge representation on consistency

| Model | Triads | Triads with merged head and tail entities | Natural language knowledge text | | |
|---|---|---|---|---|---|
| | | | no-shuffle | Shuffle:5 | Shuffle: 20 |
| Qwen2-72B | 0.67 | 0.71 | 0.73 | 0.75 | 0.76 |
| Qwen1.5-32B | 0.64 | 0.69 | 0.72 | 0.74 | 0.75 |
| GPT-3.5turbo | 0.60 | 0.62 | 0.69 | 0.71 | 0.74 |

From the experimental results, it can be seen that the consistency of the interpretations generated by the RIP4LCA system in the case of using knowledge triads is 67%, 64% and 60% on the three LLMs, respectively; and the consistency of the interpretations generated by the system in the case of using merged head and tail entity triads is 71%, 69% and 62% on the three LLMs, respectively. In the third group of experiments, we use the knowledge representation of natural language text, and we do three more sets of experiments in this group separately to assess the effectiveness of the knowledge slice shuffle method proposed in this paper.

As can be seen from Table 2, the consistency of the generated interpretations is improved using the knowledge representation of natural language text over the original triad and the merged head and tail triad representation, regardless of whether the knowledge is subjected to the shuffle operation or not. Meanwhile, the experimental results of the third group also show that shuffle operation on knowledge significantly improves the consistency of the interpretation, and, the more the number of shuffles, the higher the consistency. However, as the number of shuffles gradually increases, the system performance decreases. After experiments, we find that when the number of shuffles is set to 20, the consistency of the interpretation stabilizes, and the overall performance of the system is not significantly affected. We visualize the distribution of the consistency metrics using violin plots. The consistency distributions of the final output interpretations on the three LLMs without shuffle operation, with a shuffle count of 5 and a shuffle count of 20, are shown in Fig.5, where the Y-axis represents the consistency measure of the interpretative content, and the width of the image corresponding to each Y-value represents how many interpretative tasks have an interpretation consistency measure for this Y-value.

## 4.3 COVERAGE CALCULATION

To further evaluate the validity of the result-interpretable amyloidosis risk prediction system realized in this paper for Q&A-type tasks and to validate the coverage calculation method proposed in

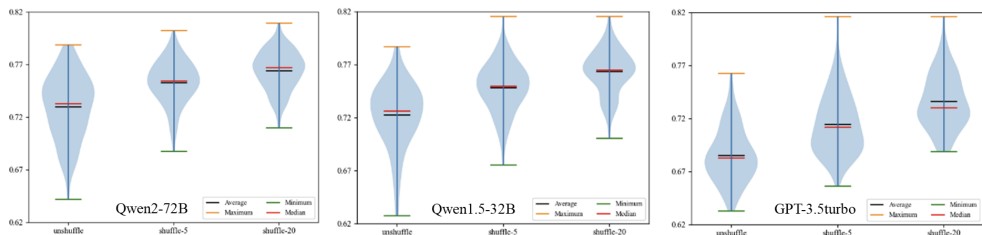

Figure 5: The consistency distributions of the interpretations on three LLMs

this paper, two groups of experiments were conducted. Each group of experiments involves 200 Q&A-type tasks, i.e., the system outputs specialized answers to 200 questions posed by hematology clinicians. The coverage of each group of experiment is taken as the average of the coverage of all tasks.

The first group of experiments inputs questions from physicians about amyloidosis diseases and generates answers directly from the large language model. The coverage calculation for this experiment is done by direct semantic similarity comparison. That is, the physician's question and the answer generated by the LLMs are semantically segmented to form a collection of question semantic slices and a collection of answer semantic slices, and then the semantic similarity algorithm is used to compare the semantic slices in the two collections for similarity, and to find out which physician's concern in question are covered by the output of the LLMs.

The second group of experiments input questions from physicians about amyloidosis diseases into our RIP4LCA system, which generates the content of the responses, but instead of using the coverage calculation method of multiple-issue similarity comparison and the coverage discriminator to select the optimal outputs, we used the same semantic similarity comparison method as in the first group of experiments for the coverage calculation.

The results of two groups of experiments are shown in Table.3. The first group of experiments using Qwen2-72B, Qwen1.5-32B and GPT-3.5turbo yielded average coverage of 0.44, 0.46 and 0.33, respectively. The second group of experiments using Qwen2-72B, Qwen1.5-32B and GPT-3.5turbo yielded average coverage of 0.68, 0.62 and 0.49, respectively. The results show that the system realized in this paper enables the answers to better cover the physician's concerns due to the identification and targeted enhancement of knowledge retrieval for the issues of the physician's questions.

To further validate the coverage calculation method based on multi-issue comparison proposed in this paper, we conducted another 6 sets of tests. These additional tests were performed entirely using RIP4LCA realized in this paper, i.e., the coverage calculation method of multi-issue similarity comparison was used and the output was optimized by a coverage discriminator with at least 3 rounds of judgment. All 6 sets of tests perform semantic segmentation on the answers generated by the interpreter invoking the LLM and generate a number of issues corresponding to each semantic slice, and the number of generated issues is 1,2,3,4,5,6 in each of the 6 sets of tests, respectively. The coverage was calculated using the method described in section 3.2 of this paper. The results are also listed in Table.3.

In set 1 additional tests, only one corresponding issue is generated for a semantic slice, and after the issue is generated, the coverage is then computed by the multi-issue similarity comparison method, and the results are basically the same as the results of the coverage computation method that direct semantic similarity comparison. As the number of issues generated by a semantic slice increase, the coverage metrics of the content of the answer rise significantly. The best results were achieved when the number of issues was 5. The number of issues to be generated actually needs to consider the influence of many aspects. Few issues cannot realize the accurate calculation of coverage. Too many issues will also have an adverse effect on the accuracy of coverage, such as introducing the error in issue generation into the process of coverage calculation, and too many issues will also cause the performance of the system to be degraded. Therefore, the number of issues generated for each semantic slice in our RIP4LCA system is set to 5.

Table 3: Experiments on coverage of interpretation

| Model | Semantic similarity comparison | | RIP4LCA with Multiple-issue comparisons | | | | | |
|---|---|---|---|---|---|---|---|---|
| | LLM directly | RIP4LCA without multiple-issue comparisons | Issue:1 | Issue:2 | Issue:3 | Issue:4 | Issue:5 | Issue:6 |
| Qwen2-72B | 0.44 | 0.68 | 0.69 | 0.77 | 0.78 | 0.81 | 0.84 | 0.82 |
| Qwen1.5-32B | 0.46 | 0.62 | 0.62 | 0.68 | 0.70 | 0.76 | 0.77 | 0.75 |
| GPT-3.5turbo | 0.33 | 0.49 | 0.49 | 0.57 | 0.58 | 0.61 | 0.62 | 0.60 |

## 4.4 PROFESSIONALISM ASSESSMENT

In order to assess the medical professionalism of the interpretations and responses generated by RIP4LCA system realized in this paper, we conducted five groups of experiments, with groups 1, 2 and 3 of the experiments being conducted on 380 interpretive tasks. Group 1 experiments generate interpretations directly for interpretive tasks using the LLMs, group 2 uses RIP4LCA but does not employ the knowledge shuffle method and group 3 uses RIP4LCA and employs the knowledge shuffle method. Groups 4 and 5 experiments were conducted on 200 Q&A-type tasks. Group 4 experiments use the LLMs directly to generate answer and group 5 generated answer content using RIP4LCA. We invited 6 clinicians specializing in hematology to assess the professionalism of the generated content of the interpretive tasks and the Q&A-type tasks in all the experiments independently. Professionalism was measured on a scale of 1 to 5, with a maximum of 5 and a minimum of 1. The average of all clinicians' scores was taken as the final professionalism score. The results of the assessment are shown in the Fig.6. As can be seen from the results, the RIP4LCA system realized in this paper shows more professional both in the interpretive tasks and Q&A-type tasks.

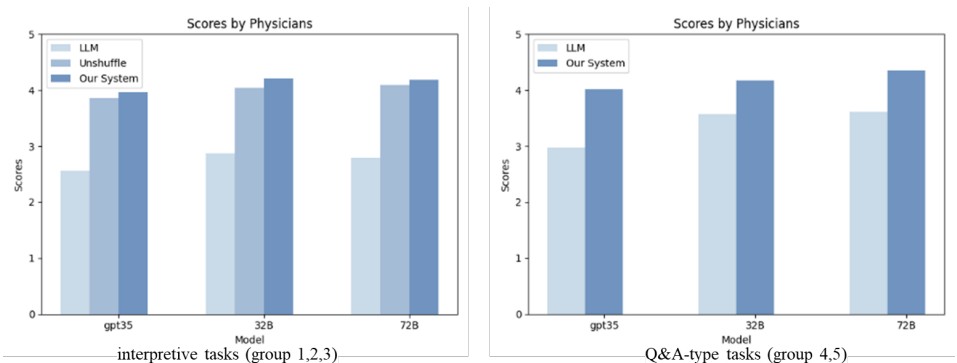

Figure 6: Professionalism assessment

## 5 CONCLUSION

This paper proposes a definition of result interpretability for medical AI, and divides explainable medical AI research into three phases: data explainability, process explainability and result inter-pretability. An architecture for result-interpretable medical AI system based on AI-Agent is also proposed and a result-interpretable amyloidosis risk prediction system is realized, which enables professional interpretation of the result of the risk prediction model for amyloidosis disease through a LLM and supports professional Q&A with clinicians. The results of the experiments show that the result-interpretable system realized in this paper performs well and is able to provide clinicians with professional, specialized interpretations and Q&A based on the medical AI results and the relevant input indicators that meet the actual needs of the clinic.

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
