# OpenReview forum: "A Research on Result Interpretability of Medical AI Based on Large Language Model"
_ICLR.cc/2025/Conference — ICLR 2025 Conference Withdrawn Submission_

### Official Review · Reviewer_1Yi5 · 2024-11-03

**Soundness:** 2
**Presentation:** 2
**Contribution:** 2
**Rating:** 5
**Confidence:** 5

**Summary:**

The paper introduces "result interpretability" as a novel approach to enhance clinician trust in medical AI outputs, focusing on interpretability of results rather than only process transparency. It proposes a comprehensive framework with three interpretability phases—data, process, and result—to streamline the application of medical AI in clinical settings. Using an amyloidosis risk prediction model, the authors design a system integrating large language models (LLMs) to support interpretative quality through interpretative integrity measures like consistency, coverage, and professionalism, which assess the completeness and clinical relevance of AI outputs. Experimentation shows that this approach increases interpretative accuracy and clinician acceptance.

**Strengths:**

The paper is an interesting attempt at solving the roadblocks to deploy medical AI in practice. Some of the key strengths are:
1. Explainaibility is a concept that has been continued to be studied for a formal definition based on their practical settings. In this paper, the authors proposed another angle by trying to  shift the focus from process interpretability to result interpretability. In their specific circumstances, they aimed to make it align more directly with clinicians' needs by enabling them to make informed decisions based on AI recommendations without fully understanding the AI's internal workings. This hypothesis is understandable and exhibits a novel contribution.
2. The authors also tried to test for the various dimensions reported in their paper by setting up an experimentation framework involving multiple LLMs and interpretative tasks. They are able to identify specific patterns that provides support for the effectiveness of the proposed interpretability framework, which enhances consistency, coverage, and professionalism in AI-driven predictions.
3. Finally, some of the metrics proposed herein such as consistency, coverage, and professionalism could be a interesting addition in making AI deployment to real world medical settings practical

**Weaknesses:**

However given the promise there are a number of ways this paper can be improved upon

1. The framework would benefit from engaging more thoroughly with existing research on clinician-driven explainability and mental models of explainability, such as those addressed in user-centered studies, to strengthen the theoretical underpinnings. Of special notes are the works that derives from Miller et al [1]. Also while the authors claim propose a novel "results explainability" related concepts have been studied in user-driven explainability, especially considering medical personas and their various needs Dey et al. [2]. Finally, the concept of using LLMs to provide holistic explanations have also been studies in literatures. Of special note are the works from Slack et al [3]
2. While experimental validation is robust, real-world clinical testing or feedback on usability and effectiveness from clinicians would bolster the practical implications of this framework. Especially it is important to show whether the various dimensions transformed to how it would improve the time to deployment or overall improvement in clinical workflow.
3. Finally, the study is based on a rare disease (amyloidosis), which may limit the generalizability of the interpretability framework to other, potentially less specialized, medical domains.


[1]: Miller et al, https://www.sciencedirect.com/science/article/pii/S0004370218305988.

[2]: Dey et al; https://www.cell.com/patterns/fulltext/S2666-3899(22)00078-2.

[3]: Slack et al; https://www.nature.com/articles/s42256-023-00692-8.

**Questions:**

See weakness above. Especially it would be useful to note how the authors distinguish their contribution from user-driven explainability method.

Also, it is interesting to see the results but it is wholly not surprising to see RAG based models performed better than non RAG systems. What is more important is perhaps how the prompts were generated. Did the authors have to employ specific prompt engineering? Also how would this scale to other use cases

---

### Official Review · Reviewer_njoK · 2024-11-04

**Soundness:** 3
**Presentation:** 3
**Contribution:** 3
**Rating:** 8
**Confidence:** 4

**Summary:**

The authors describe how an AI agent was built to provide interpretable results regarding disease risk prediction. In particular, they propose an architecture and describe an implementation along with the results obtained for a real-world use case concerning amyloidosis.

**Strengths:**

- The authors tackle a relevant real-world problem
- Most of the content is well structured and clearly presented.
- The authors explore the intersection of explainability and LLMs - a hot topic right now

**Weaknesses:**

- We consider the authors should improve the related work. In particular, we found it has very few references to the scientific literature. We provide a detailed list of suggested works in the questions section.
- The authors provide a Figure detailing the system for amyloidosis, but they do not elaborate in detail in the manuscript. Therefore, it is unclear whether all of the components were implemented or if they provide a conceptual diagram explaining how the current research fits within a larger research picture.
- It is unclear how the prediction model is used to elaborate downstream explanations along with the results obtained from the RAG.
- The authors should assess whether the models' differences are statistically significant.

**Questions:**

Dear authors,

We consider the research relevant and interesting. Nevertheless, we would like to point to certain improvement opportunities:

GENERAL COMMENTS

(1) - While the related work is mostly well-written, we consider it has very few references to scientific literature and little attention is devoted to the use of LLMs for the purpose of explainability - one of the key components of this research. We encourage the authors to strengthen it referencing surveys and works specific to explainability, LLMs and explainability, and explainability in the medical domain. Here some works that could be relevant to the authors: (a) Arrieta, Alejandro Barredo, et al. "Explainable Artificial Intelligence (XAI): Concepts, taxonomies, opportunities and challenges toward responsible AI." Information fusion 58 (2020): 82-115., (b) Adadi, Amina, and Mohammed Berrada. "Peeking inside the black-box: a survey on explainable artificial intelligence (XAI)." IEEE access 6 (2018): 52138-52160., (c) Tjoa, Erico, and Cuntai Guan. "A survey on explainable artificial intelligence (xai): Toward medical xai." IEEE transactions on neural networks and learning systems 32.11 (2020): 4793-4813., (d) Minh, Dang, et al. "Explainable artificial intelligence: a comprehensive review." Artificial Intelligence Review (2022): 1-66., and (e) Cambria, Erik, et al. "XAI meets LLMs: A Survey of the Relation between Explainable AI and Large Language Models." arXiv preprint arXiv:2407.15248 (2024).

(2) - The authors use a RAG architecture to implement an interactive system to follow up on physicians' questions. We encourage the authors to include at least some references to this field, architectures, challenges, and their relationship to XAI. The authors may be interested on the following works: (a) Fan, Wenqi, et al. "A survey on rag meeting llms: Towards retrieval-augmented large language models." Proceedings of the 30th ACM SIGKDD Conference on Knowledge Discovery and Data Mining. 2024., (b) Gao, Yunfan, et al. "Retrieval-augmented generation for large language models: A survey." arXiv preprint arXiv:2312.10997 (2023)., (c) Zhao, Siyun, et al. "Retrieval Augmented Generation (RAG) and Beyond: A Comprehensive Survey on How to Make your LLMs use External Data More Wisely." arXiv preprint arXiv:2409.14924 (2024)., (d) Wu, Xuansheng, et al. "Usable XAI: 10 strategies towards exploiting explainability in the LLM era." arXiv preprint arXiv:2403.08946 (2024)., (e) Tekkesinoglu, Sule, and Lars Kunze. "From Feature Importance to Natural Language Explanations Using LLMs with RAG." arXiv preprint arXiv:2407.20990 (2024)., and (f) Schneider, Johannes. "Explainable generative AI (GenXAI): A survey, conceptualization, and research agenda." Artificial Intelligence Review 57.11 (2024): 289.

(3) - "The measure of professionalism was scored on a scale [...] given independently by 6 different clinicians" -> we would appreciate some summary statistics/distribution of the scores given by each clinician, to understand e.g., if they rated optimistically or not. (a) did the authors normalize the scores for each clinician?, (b) could the authors provide some additional information regarding the clinicians? E.g., what is their seniority and age?

(4) We encourage the authors to conduct an assessment of whether the differences among the models' results are statistically different. For example, in Table 1, for the different LLM models used within Group 2, does using a different LLM result in statistically significant better outcomes?

(5) - In Fig. 2, the authors mention many modules that are not explained in detail in the manuscript. E.g., (a) how is reinforcement learning with human feedback implemented and used in this research?, (b) how do the authors perform hallucination control?, (c) what is the purpose and how is the intention recognition used?

(6) - It is unclear to us what kind of information from the prediction model is used to feed to the interpreter and how these are joined to the RAG outputs. Furthermore, are these requests done in parallel, or is there a specific order to be followed (e.g., to constrain the RAG search)? Are Figures 1 and 2 related?


FIGURES

(7) - Figure 2, 3, 4: Enhance the captions. Why are LLM-related components highlighted? For example, in Fig. 4, why do the authors highlight the "Issue vector" box?

TABLES
(8)- All tables reporting metrics: (a) provide arrows next to the metric names (up/down) indicating whether a higher/lower result is better; (b) bold the best results; (c) align numbers to the right, to make differences of magnitude evident to the reader.

MINOR COMMENTS

(9)—The authors mention that "shuffle(K,i) denotes the ith randomized sorting operation on the knowledge slices set K." This is a common practice—is there perhaps some literature supporting it?

(10)- We encourage the authors to revise the manuscript for typos and to fix them.

---

### Official Review · Reviewer_VUgC · 2024-11-04

**Soundness:** 2
**Presentation:** 3
**Contribution:** 2
**Rating:** 3
**Confidence:** 3

**Summary:**

The paper presents the notion of results interpretability for the medical AI models and presents a framework based on LLMs and knowledge graphs to provide interactive explanations to the users (i.e., health care providers). The aim is to provide consistent, reliable explanations to induce trust in the users of the black-box machine learning models.

**Strengths:**

This paper makes a to the field of medical AI by introducing the concept of result interpretability—focusing on providing interpretable, clinically relevant outputs that enhance understanding and usability in medical decision-making contexts. The authors also introduce interpretative integrity, a structured approach to measure the effectiveness of result interpretations using three key metrics: consistency, coverage, and professionalism. This focus on end-user interpretability, alongside a practical framework for assessing interpretative quality, helps to bridge the gap between model performance and clinical trust.

**Weaknesses:**

Major:
1) Risk of Hallucination in LLMs: While the paper demonstrates the use of large language models (LLMs) for generating interpretative outputs, it overlooks the significant risk of LLM hallucinations—instances where the model might produce inaccurate or fabricated information. The use of RAG could diminish the risk, but it does not obviate it - the hallucination of LLMs is a direct paradox to the claims the authors made on inducing trust in the users of black-box machine learning models. This limitation could compromise the reliability of the explanations in clinical settings, where accuracy is critical.

2) Alternative Knowledge Graph-Based Explainability Methods: The paper’s approach to result interpretability seems to be a straightforward extension of existing explainability methods that use knowledge graphs to enhance model interpretability (see [1] for a reference on the literature in this domain). Other methods have already demonstrated effective use of knowledge graphs to produce clinically relevant explanations. Compared to feature attribution methods (like SHAP), these graph-based approaches often provide more domain-specific insights, bridging gaps in feature-level explanation limitations. Consequently, the proposed "result interpretability" framework appears less innovative and more like an auxiliary use of LLMs with KG-based explanation, adding little to the existing knowledge graph explainability frameworks.

Minor:
A main subtle assumption in the proposed framework is the existence of a complete and accurate KG, which is not accessible most of the time; this should have been discussed as a serious limitation of the work;

[1]Tiddi, Ilaria, and Stefan Schlobach. "Knowledge graphs as tools for explainable machine learning: A survey." Artificial Intelligence 302 (2022): 103627.

**Questions:**

Aside from above concerns, I have a question about the consistency check. The system incorporates a consistency check using cosine similarity with factual knowledge; does this metric alone fully capture inconsistencies or subtle inaccuracies in the interpretations? A simple cosine similarity measure does not seem to catch nuanced or misleading language generated by the LLM, and this might potentially affect reliability.

---

### Official Review · Reviewer_kMst · 2024-11-04

**Soundness:** 3
**Presentation:** 2
**Contribution:** 1
**Rating:** 3
**Confidence:** 4

**Summary:**

This paper introduces an architecture for a result-interpretable medical AI system, centered on a large language model (LLM) designed to enhance interpretability in risk prediction for amyloidosis. Through experiments involving six professional clinicians, the study demonstrates that the proposed system significantly improves the acceptance of AI models in medical diagnosis support.

Although the case study offers valuable insights, the methodological contributions appear limited. Additionally, the experimentation is constrained to a single medical case study, without comparisons to alternative approaches aside from the original LLM.

**Strengths:**

the case study offers valuable insights

**Weaknesses:**

the methodological contributions appear limited. Additionally, the experimentation is constrained to a single medical case study, without comparisons to alternative approaches aside from the original LLM.

**Questions:**

Have you considered adapting existing methods that could be easily modified to perform the same tasks as RIP4LCA?

Additionally, is your proposed approach applicable solely to the case of amyloidosis?

---

### Note · Authors · 2024-11-20

I have read and agree with the venue's withdrawal policy on behalf of myself and my co-authors.